# Pancreatico-Jejunostomy Fistula After Pancreaticoduodenectomy: Where Do We Stand? Results from an International Survey

**DOI:** 10.3390/curroncol32120657

**Published:** 2025-11-24

**Authors:** Silvio Caringi, Michele Tedeschi, Antonella Delvecchio, Annachiara Casella, Valentina Ferraro, Cataldo De Palma, Rosalinda Filippo, Matteo Stasi, Tommaso Maria Manzia, Riccardo Memeo

**Affiliations:** 1Unit of Hepato-Biliary and Pancreatic Surgery, “F. Miulli” General Hospital, Acquaviva delle Fonti, 70021 Bari, Italy; m.tedeschi@miulli.it (M.T.); a.delvecchio@miulli.it (A.D.); a.casella@miulli.it (A.C.); v.ferraro@miulli.it (V.F.); r.filippo@miulli.it (R.F.); matteo.stasi@miulli.it (M.S.); r.memeo@miulli.it (R.M.); 2Department of Surgery, Università Degli Studi Roma “Tor Vergata”, Via Montpellier 1, 00133 Rome, Italy; 3Department of Medicine and Surgery, LUM University, Casamassima, 70010 Bari, Italy; 4IRCCS Humanitas Research Hospital, Pancreatic Surgery, Via Alessandro Manzoni 56, 20089 Rozzano, Italy; cataldo.depalma@st.hunimed.eu; 5Transplant and HPB Unit, Department of Surgery Sciences, University of Rome Tor Vergata, 00133 Rome, Italy; manzia@med.uniroma2.it

**Keywords:** pancreatic surgery, pancreatic fistula, pancreatic anastomoses, pancreatic stent

## Abstract

Pancreaticoduodenectomy (PD) is among the most challenging abdominal operations, and postoperative pancreatic fistula (POPF) is its worst complication. This international survey was answered by 122 participating pancreatic surgery units in 26 nations to find current trends in pancreatico-jejunostomy (PJ) reconstruction and pancreatic stenting practice. Most of the units performed an anastomosis between the mucosa and duct and routinely employed stents. High-volume centers and those with routine stenting had reduced mean rates of clinically significant POPF, but the PJ technique itself did not significantly affect fistula formation. These findings suggest that institutional surgical volume and perioperative standardization are more important in the determination of postoperative results than technical modifications per se. International centralization and procedural standardization initiatives are therefore imperative to reducing severe POPF rates and improving the safety of pancreatic surgery.

## 1. Introduction

Pancreaticoduodenectomy (PD) is still the first choice for benign and malignant pancreatic head and periampullary area tumors. It is a complex procedure with wide resection and at least three anastomoses based on the reconstruction technique. There have been numerous techniques described for these anastomoses, with preference more commonly being surgeon experience and institutional preference. Recent comprehensive reviews further summarize contemporary techniques and their relative advantages and limitations [1].

Among the various techniques available, pancreatico-jejunostomy (PJ), an anastomosis between the pancreatic remnant and a loop of jejunum, has emerged as the most widely used reconstruction [2]. Despite its technical difficulty, mortality after PD has declined to less than 5% in high-volume experienced centers [3,4]. Morbidity remains high, and postoperative pancreatic fistula (POPF) remains the most dreaded complication, with rates reported between 5% and 40% [5].

Furthermore, the development of POPF is frequently linked to the technical failure of the PJ. Several intraoperative factors contribute to this risk, including the width of the pancreatic duct, pancreatic gland texture, vascularity of the stump, method of suture placement, and tension over the anastomosis. Technical refinement alone, however, has not eliminated POPF, suggesting that surgeon experience, institutional volume, and perioperative management also play crucial roles.

Unlike previous national or regional surveys, this study provides a contemporary global view of PJ operations on multiple continents, addressing shifting trends in surgical technique, stent use, and institution centralization following the updating of the 2016 ISGPS definition. To our best knowledge, it is one of the most geographically broad and recent surveys on the PJ technique and stent usage, with the intention to inform global practice and direct priorities for future research.

## 2. Materials and Methods

A web-based questionnaire was distributed between December 2024 and February 2025 via professional networks and the ASHBPS mailing list. The survey had nine multiple-choice and open-ended questions (Table 1).

The survey was developed by the principal research team and internally reviewed. No pilot testing or formal validation of the questionnaire was performed before dissemination, which may affect the reliability and interpretability of the responses. Key clinical predictors of POPF, such as pancreatic duct diameter, gland texture, pathology, or intraoperative blood loss, were not collected. Therefore, risk stratification using validated scores (e.g., Fistula Risk Score) was not possible. Invites went out through professional networks and society mailing lists; because lists were copied, an exact response denominator could not be ascertained. Duplicate institutional responses were kept to a minimum through cross-referencing all incoming surveys for duplicate institutional identifiers and e-mail addresses; duplicate submissions were deleted.

All data were collected and verified centrally in March 2025. To avoid duplicate entries, each center was required to indicate the institution name and country. Incomplete or duplicate entries were manually screened and excluded. POPF grades B and C were classified according to the 2016 ISGPS criteria [6]. Answers were classified into analysis groups as follows:Centers performing <50 vs. ≥50 PDs per year. The 50 PDs per year cutoff was chosen based on prior volume–outcome studies that have used similar cutoffs to distinguish high- versus low-volume centers. While definitions vary across studies, a 50-case cutoff is an appropriate differentiation for subgroup analysis in our population.Centers performing duct-to-mucosa vs. other PJ techniques.Centers performing vs. not performing stents in the PJ.

Mean values and standard deviations for POPF grades B and C in both groups were calculated. The Student’s *t*-test (two-tailed, unequal variance) was applied to compare. A *p* < 0.05 was used to denote statistical significance.

Given the nature of the dataset and the absence of key covariates, multivariable regression modeling was not feasible.

It was not felt that ethical approval was needed, as no patient-identifiable data had been collected. The research was conducted in accordance with the Declaration of Helsinki principles.

## 3. Results

A total of 122 pancreatic surgical units in 26 countries on five continents responded to the survey, providing a broad view of modern international practice. There was a widespread response from European centers, with some returned from Asia, Oceania, and the Americas (Table 2).

The centers were intentionally anonymized, and the subgroup statistical comparisons were descriptive and not powered for visual effect.

The majority of responding centers reported a relatively low annual pancreaticoduodenectomy (PD) volume, a measure of global variation in the distribution of cases and institutional specialization. A smaller but noteworthy proportion of high-volume centers contributed, so meaningful comparison was possible across different levels of activity. Minimally invasive approaches (laparoscopic and robotic) are increasingly adopted worldwide; however, in the responding centers, open pancreaticoduodenectomy remains the predominant approach in most units. In our sample, only 21% of centers performing minimally invasive PD reported that >50% of their PDs were robotic, and only 4.9% reported >50% laparoscopic PDs, indicating that open PD is still the most frequent operative pathway in everyday practice (Table 3).

For pancreatico-jejunostomy (PJ) reconstruction, duct-to-mucosa anastomosis was favored by the overwhelming majority of respondents as the clear favorite. Very few respondents had used invagination or other modified methods, showing a global trend toward the adoption of the duct-to-mucosa design as a standard of care.

All but one center employed a stent on a routine basis in the PJ, either universally or selectively based on the patient’s fistula risk score. The most commonly used device was the plastic or silicone feeding tube, and commercially available stents were less so. The variability indicates the persistent lack of consensus regarding stent material and management options.

On comparison of postoperative outcomes, the postoperative pancreatic fistula (POPF) rates clinically significantly varied institutionally but revealed consistent patterns. Institutions of high volume had significantly reduced rates of severe (grade C) fistula, supporting the evidence for the relationship between procedural centralization and improved surgical outcomes. The PJ anastomosis type itself had no meaningful effect on the POPF rates, confirming that technique selection will not determine fistula risk alone.

Although stent use was not statistically significant, positive trends toward diminished rates of POPF were observed in centers where routine stenting was embraced. This suggests the potential for a protective effect to develop in larger stratified analyses. Together, these findings reinforce anew that surgical volume and postoperative care practices matter more in POPF outcomes than some differences in anastomosis.

Subgroup analysis led to the results shown in Table 4.

We performed a focused description of centers performing ≥ 50 PD/year (n = 30, 24.6% of respondents). In this subset, duct-to-mucosa anastomosis remained the predominant technique (>80% of high-volume centers), and routine stent placement was reported by the majority. Mean reported rates of clinically relevant POPF (grades B–C) in these high-volume centers were lower for grade C fistula (mean 3.95% ± 2.39) compared with low-volume centers (mean 6.47% ± 8.52; *p* = 0.013). The pattern of technique selection in high-volume centers did not substantially differ from the overall cohort; however, high-volume centers tended to report more standardized perioperative pathways and selective use of stents in higher-risk glands (as self-reported).

## 4. Discussion

Pancreaticoduodenectomy (PD) remains the most complex and hazardous of abdominal surgeries. Notwithstanding refinements over decades, postoperative pancreatic fistula (POPF) continues to be the major determinant of morbidity, longer hospital stay, and mortality after PD. The present global survey gives a snapshot impression of international practice for pancreatico-jejunostomy (PJ) reconstruction and the use of pancreatic duct stent, with data collected from 122 units in 26 countries. Its findings highlight the ongoing popularity of duct-to-mucosa anastomosis and stenting, but more importantly, endorse the growing recognition that institutional volume and compliance with standardized perioperative protocols have a greater impact on results than any technical variation in its own right.

The observation that POPF persists in the face of overwhelming technical progress highlights its multifactorial etiology. Whereas the 2016 ISGPS update classified and graded its definition on standardized criteria [6], it also noted that POPF is caused by the multifactorial interplay of gland texture, ductal size, perfusion, and surrounding enzymatic damage. Postoperative acute pancreatitis has also been identified as an independent predictor of clinically relevant POPF and should be considered among recognized risk factors [7]. There have been various efforts at risk reduction by optimizing anastomotic technique—ranging from duct-to-mucosa and invagination to hybrid and modified Blumgart techniques [8,9,10,11]. However, according to our findings, no method has appeared to be consistently superior. Randomized trials and meta-analyses demonstrate the equivalence of clinically significant POPF rates among these procedures when conducted by experienced surgeons in well-organized programs [12,13,14]. Thus, the question has more and more become “under what conditions and in what context can technique be standardized safely”, and less one of “which technique is best”.

Duct-to-mucosa anastomosis is the global preference, primarily due to its natural jejunal and pancreatic epithelial alignment, measured approximation, and easy integration of the stent. Nonetheless, technical precision is conceded by its authors to be less decisive than those variables associated with the patient or the institution. Duct diameter and gland texture are still significant risk factors; minimal duct, non-fibrotic, and soft pancreas are consistently associated with higher POPF occurrence [15,16]. These anatomic properties, as a result of anatomical design, are usually more critical than the impact of anastomotic design itself. Moreover, perioperative fluid resuscitation, use of somatostatin analogues, drain policy, and postoperative monitoring have huge institutional variability and contribute heavily to outcome heterogeneity [17,18]. This multifactorial etiology is the very reason why multicenter trials frequently cannot show statistically significant differences between anastomotic types.

Our results reinforce the known relationship between institutional case volume and POPF outcome. Units with ≥50 PDs/year had fewer grade C fistulas than low-volume centers, a finding in line with various volume–outcome studies [19,20,21]. High-volume facilities include those with specialized teams, multidisciplinary coordination of care, audit-performed routines, and evidence-based algorithm adherence—factors that all favor procedure standardization and effective complication detection. The experience factor goes beyond individual technical competence; it includes team experience with aggressive perioperative care, including early sepsis treatment, nutritional optimization, and aggressive intervention in developing fistulas. Centralization of pancreatic resections is therefore a quality imperative. National initiatives in Europe and Asia demonstrate that offloading PD procedures to high-volume centers realizes measurable decreases in mortality and major morbidity without decreasing access [22,23]. However, without multivariable adjustment, potential confounders, such as case complexity, may explain differences observed between high- and low-volume institutions.

Pancreatic stenting is another topic of long-standing controversy. Developed initially to divert pancreatic juice away from anastomosis and reduce intraductal pressure, stents have been employed in numerous configurations as internal or external drainage systems. In our survey, nearly four-fifths of centers reported the routine use of stenting, most often with plain plastic feeding tubes rather than proprietary commercial devices. Although meta-analyses have reported modest decreases in clinically relevant POPF with stenting, results remain inconsistent [24,25,26,27]. The pathophysiologic mechanism of possible protection—mechanical decompression and targeted ductal drainage—is biologically plausible, but clouded by heterogeneity of stent type, size, site, and timing of removal. External stents reduce fistula rates but are uncomfortable and lead to infection; internal stents are well-tolerated but can migrate or become obstructed. Furthermore, the stenting benefit would be greatest in risky glands (soft consistency, small duct) and is not always effective [28]. Its selective action most likely explains the bias toward risk-stratified, individualized stenting rather than unselective application.

Other than technical and device-related factors, POPF incidence is regulated by intraoperative management and perioperative environment. Callery et al.’s development of the Fistula Risk Score (FRS) provided a validated basis for pre-emptive risk stratification [15]. Through the integration of duct size, gland texture, pathology, and blood loss, the FRS makes possible the institution of targeted preventive interventions, including anastomosis choice, stent use, and postoperative follow-up intensity. Implementation of the FRS into practice has been associated with the improved detection and treatment of high-risk circumstances [29,30,31]. Some volume-high institutions now incorporate this scoring system into operative planning, demonstrating the shift from intuition-guided to evidence-based surgical decision-making.

The advent of minimally invasive pancreaticoduodenectomy (MIPD)—laparoscopic and robotic—is superimposed. In spite of encouraging descriptions of equal oncologic outcomes and reduced wound morbidity, MIPD remains technically demanding and learning curve-sensitive. A few registries and meta-analyses have documented increased risk of POPF with early learning curves [32,33,34]. While at proficiency, outcomes are as good as, or superior to, those of open PD, and particularly with standardized reconstruction and team experience maintained. Our survey’s findings that nearly half of the respondents performed some percentage of PDs robotically prove the general availability of the technologies and necessitate formalized training programs and credentialing systems to allow for safe implementation.

Institutional policies for the management of drains have also evolved. The customary intra-abdominal drainage after PD has been questioned by randomized controlled trials, which have shown that omitting drains in low-risk patients will not lead to increased morbidity [35,36]. However, in high-risk glands, drains remain an absolute requisite for the early detection of biochemical leaks and the prevention of undrained collections that can trigger sepsis. The timing of drain removal is also crucial; early removal (postoperative day 3) has also been associated with fewer infectious complications in the absence of biochemical POPF [37]. Our findings cannot be applied to guide drain strategies but align with evidence for individualized, risk-directed strategies.

The use of somatostatin analogues for the prevention of POPF is still controversial. Initial meta-analyses were positive, but subsequent randomized controlled trials and network analyses indicated variable or negligible benefits [37,38]. The new PREVFAIL trial (2024) could not demonstrate a statistically significant reduction in grade B/C POPF with routine octreotide administration [39]. Guidelines, therefore, now recommend selective, rather than routine, prophylaxis. Similarly, intraoperative topical sealants and biological glues—though theoretically attractive—are devoid of uniform efficacy evidence [40,41].

One highly underappreciated predictor of outcome is perioperative standardization. Data from improved recovery after surgery (ERAS) protocols demonstrate that systematic pathways such as nutrition, early mobilization, and multimodal analgesia radically reduce hospital stay without introducing complications [42,43]. When ERAS is incorporated with standardized surgical technique and postoperative care algorithms, overall improvements in safety and efficiency emerge. Low variance in grade C POPF among high-volume, standardized institutions in our survey demonstrates this principle.

We note that, although the drainage policy, the use of somatostatin analogues, and the ERAS protocols are important determinants of postoperative course and are discussed here for context, these items were not included in our questionnaire, and therefore, no inferences can be drawn from our data regarding their association with POPF incidence.

Our findings also uncovered significant heterogeneity of practice worldwide. An estimated 75% of responding centers were European, reflecting an imbalance in keeping with the regional concentration of expertise in pancreatic surgery but limiting extrapolation to disadvantaged continents. Heterogeneity of stent material, volume thresholds for surgery, and perioperative care reflects the absence of concordance in the face of several decades of research. However, the survey does provide useful insight into contemporary trends and potential for harmonization. The establishment of international registries and quality collaboratives, such as the European Registry for Pancreatic Surgery (EUREPA) and the American College of Surgeons NSQIP Pancreatectomy Module, offers an opportunity for benchmarking and ongoing improvement [44].

## 5. Limitations

There are a number of limitations in this study. Firstly, the survey design relied on self-reported institutional data that are subject to reporting bias and accuracy limits. Respondents can underestimate or overestimate POPF rates or modify answers to reflect perceived best practice.

Secondly, the voluntary participation resulted in a selection bias towards more academically collaborative centers or those with an interest in pancreatic surgery outcomes.

Third, the use of a limited number of multiple-choice questions restricted analysis depth; valuable parameters such as pancreatic duct diameter, texture of the gland, and underlying pathology were not collected.

In addition, three-fourths of the centers participating in the survey were European, limiting the generalizability of our findings to underrepresented areas in the survey. Furthermore, institutional rates derived from self-reporting are susceptible to recall and reporting bias, and differences in local practice/definitions can influence the comparability of the POPF rates reported.

Furthermore, the sample was highly unbalanced, with only 24.6% of centers performing ≥ 50 PD/year. This uneven distribution reduced the statistical power and increased the risk of type II errors.

Finally, since the survey was cross-sectional, causality between surgical practice and clinical outcome could not be ascertained. Because of this, causality could not be inferred. The association between institutional volume and POPF does not prove that centralization reduces POPF; rather, it identifies a correlation.

Despite these limitations, extensive geographic representation and the number of responding centers provide a valuable global snapshot of current practice trends and patterns in PJ reconstruction.

## 6. Conclusions

Postoperative pancreatic fistula remains a significant cause of morbidity after pancreaticoduodenectomy. In this international survey, no single pancreatico-jejunal anastomotic technique emerged as universally superior in routine practice; rather, institutional case volume and organizational factors were associated with lower rates of severe (grade C) POPF in our dataset.

Our international survey demonstrates that duct-to-mucosa PJ is the most popular technique worldwide, and stent placement in the PJ is standard practice. While both stent placement and high surgical volume are correlated with lower mean POPF rates, only institutional volume achieved statistical significance for the prevention of grade C POPF.

These findings emphasize the importance of centralizing PD procedures in high-volume centers and fostering procedural standardization to improve outcomes. Further multicenter prospective studies and randomized trials are needed to validate the potential benefits of PJ stenting and continue to develop strategies for POPF prevention.

## Figures and Tables

**Table 1 curroncol-32-00657-t001:** Nine-item questionnaire.

Questions	Answers
How many pancreaticoduodenectomies do you perform in a year?	<20 21–50 51–100 >100
How many of these are performed robotically?	<25% 25–50% 51–75% >75% No one
How many of these are performed laparoscopically?	<25% 25–50% 51–75% >75% No one
Which type of pancreatico-jejunostomy do you perform?	Invagination technique Modified invagination technique (Peng, Chen’s U-suture, etc.) Duct-to-mucosa (Blumgart, Kakita, Cattell-Warren) Other (Specify)
Do you place a stent in pancreatico-jejunostomy?	Yes No
If yes, when do you place a stent?	Routinely Based on FRS Other (Specify)
If yes, which stent do you use?	Archimedes PankreaPlus Pediatric PVC feeding tube Sumitomo Bakelite’s tube Temporary external stented drainage sec. Witzel Wirsungostomy Prophylactic plastic stent Other (Specify)
What is the rate of grade B pancreatic fistulas in your center?	Essay
What is the rate of grade C pancreatic fistulas in your center?	Essay

**Table 2 curroncol-32-00657-t002:** Center distribution.

Location	*n* = 122
Europe	92 (75.4%)
Asia	15 (12.3%)
Oceania	7 (5.7%)
America	4 (3.3%)
Africa	4 (3.3%)

**Table 3 curroncol-32-00657-t003:** Surgical approach.

Frequency of Surgical Approach Performed	*n* (%)
Robotic	60 (49.2%)
<25% 25–50% 51–75% >75%	39 (65%) 8 (13.3%) 7 (11.7%) 6 (10%)
Laparoscopic	41 (33.6%)
<25% 25–50% 51–75% >75%	33 (80.5%) 6 (14.6%) 2 (4.9%) 0
Exclusively open	48 (39.3%)
Both laparoscopic and robotic	27 (22.1%)

**Table 4 curroncol-32-00657-t004:** POPF subgroup analysis. SD: Standard Deviation.

	N (%)	POPF B		POPF C	
**Volume**		**Mean (%) ± SD**	***p*-Value**	**Mean (%) ± SD**	***p*-Value**
<50 PDs/year >50 PDs/year	92 (75.4%) 30 (24.6%)	17.73 ± 10.23 15.24 ± 7.29	0.169	6.47 ± 8.52 3.95 ± 2.39	0.013
**Pancreatico-jejunostomy**					
Duct-to-mucosa Others	108 (88.5%) 14 (11.5%)	17.1 ± 9.33 17.29 ± 16.74	0.969	5.84 ± 7.81 5.75 ± 4.38	0.946
**Stent placement**					
Yes No	96 (78.69%) 26 (21.31%)	16.25 ± 8.7 20.42 ± 14.89	0.191	5.37 ± 7.49 7.58 ± 7.31	0.188

## Data Availability

The data supporting the findings of this study are not publicly available due to institutional privacy restrictions and the confidential nature of the participating centers’ responses. An anonymized version of the dataset may be provided by the corresponding author upon reasonable request for academic and non-commercial purposes.

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
