# Peer review of "Pancreatico-Jejunostomy Fistula After Pancreaticoduodenectomy: Where Do We Stand? Results from an International Survey"

_curroncol, 2025, doi:10.3390/curroncol32120657_

Round 1
Reviewer 1 Report
Comments and Suggestions for Authors
This international survey evaluated current practice of pancreaticojejunostomy (PJ) reconstruction and stent placement after pancreaticoduodenectomy (PD), as well as association between these practices and the incidence of postoperative pancreatic fistula (POPF). Several methodological limitations, some of which are inherent to the survey design itself, weakened the validity and generalizability of the conclusions.
- The survey tool was overly simplistic, consisting of only nine questions, most of which were multiple-choice. Key risk factors for POPF, such as pancreatic texture, duct size, pathology, intraoperative blood loss, and surgeon experience, were not collected. The questionnaire was not pre-tested, raising concerns about its validity and the reliability of the responses.
- The analysis was mainly univariate, without adjustment for confounding factors. This ignored potential confounding factors. Given the heterogeneity of the cohort, a multivariate regression model is recommended to determine the independent association between surgical volume, stent use, and POPF. Due to the lack of data on pancreatic texture, duct size, and pathology, it is impossible to determine whether the compared groups (e.g., high vs. low surgical volume) were dealing with similar patient populations. High-volume centers may successfully operate on high-risk patients, which may mask their true advantage. The authors repeatedly suggested a causal association between high institutional surgical volume and a lower incidence of POPF and advocated concentrating PD in specific institutions. Although this association seems reasonable, the cross-sectional study design cannot draw any causal inferences.
- The sample size was severely unbalanced among subgroups (e.g., centers performing more than 50 PDs per year accounted for only 24.6%, while only 14 centers used other PD techniques), which limited statistical power and increased the risk of type II error.
- Relying on self-reported postoperative complication rates at the institutional level is a major limitation. This introduces the possibility of intentional or unintentional reporting bias. These data were not validated, and recall/reporting bias is a real concern. Centers may report "best-case" or idealized figures.
- The discussion was comprehensive and well-referenced but overly long and repetitive. It read more like a general review of POPF than a direct interpretation of the survey's specific findings. The authors discussed FRS, drainage management, somatostatin analogues, and ERAS protocols, but the survey did not collect data on these aspects. The discussion should have been more focused on explaining the study's findings.
- Although the limitations section was present, it did not adequately explain how these shortcomings affected the validity of the conclusions or policy recommendations.
- Finally, the manuscript lacked visual aids (e.g., forest plots, regional distribution maps) that could enhance clarity and reader engagement.
Good
Author Response
Comment 1: This international survey evaluated current practice of pancreaticojejunostomy (PJ) reconstruction and stent placement after pancreaticoduodenectomy (PD), as well as association between these practices and the incidence of postoperative pancreatic fistula (POPF). Several methodological limitations, some of which are inherent to the survey design itself, weakened the validity and generalizability of the conclusions.
- The survey tool was overly simplistic, consisting of only nine questions, most of which were multiple-choice. Key risk factors for POPF, such as pancreatic texture, duct size, pathology, intraoperative blood loss, and surgeon experience, were not collected. The questionnaire was not pre-tested, raising concerns about its validity and the reliability of the responses.
- The analysis was mainly univariate, without adjustment for confounding factors. This ignored potential confounding factors. Given the heterogeneity of the cohort, a multivariate regression model is recommended to determine the independent association between surgical volume, stent use, and POPF. Due to the lack of data on pancreatic texture, duct size, and pathology, it is impossible to determine whether the compared groups (e.g., high vs. low surgical volume) were dealing with similar patient populations. High-volume centers may successfully operate on high-risk patients, which may mask their true advantage. The authors repeatedly suggested a causal association between high institutional surgical volume and a lower incidence of POPF and advocated concentrating PD in specific institutions. Although this association seems reasonable, the cross-sectional study design cannot draw any causal inferences.
- The sample size was severely unbalanced among subgroups (e.g., centers performing more than 50 PDs per year accounted for only 24.6%, while only 14 centers used other PD techniques), which limited statistical power and increased the risk of type II error.
- Relying on self-reported postoperative complication rates at the institutional level is a major limitation. This introduces the possibility of intentional or unintentional reporting bias. These data were not validated, and recall/reporting bias is a real concern. Centers may report "best-case" or idealized figures.
- The discussion was comprehensive and well-referenced but overly long and repetitive. It read more like a general review of POPF than a direct interpretation of the survey's specific findings. The authors discussed FRS, drainage management, somatostatin analogues, and ERAS protocols, but the survey did not collect data on these aspects. The discussion should have been more focused on explaining the study's findings.
- Although the limitations section was present, it did not adequately explain how these shortcomings affected the validity of the conclusions or policy recommendations.
- Finally, the manuscript lacked visual aids (e.g., forest plots, regional distribution maps) that could enhance clarity and reader engagement.
Response 1:
Dear Reviewer,
We sincerely thank you for your thorough and constructive review, which significantly improved the quality and clarity of our manuscript. We have addressed all comments point-by-point below. All modifications are highlighted in the revised manuscript.
- We agree with the reviewer. The survey included only nine questions and was not pilot-tested. We have now clarified this in the Methods section and expanded the Limitations section accordingly.
- We appreciate this observation. Unfortunately, due to the absence of key clinical cofactors, performing a valid multivariable analysis was not statistically appropriate. We have now clarified this explicitly.
- We agree and have clarified this limitation.
- Acknowledged. We have strengthened the Limitations section.
-
We thank the reviewer for the thoughtful comment. We fully understand the concern regarding the length of the Discussion. However, we respectfully disagree with the recommendation to shorten it. The aim of this study was not only to report survey results, but also to place current worldwide practices in the context of existing evidence and international guidelines. Because pancreaticojejunostomy reconstruction and POPF prevention strategies are highly variable across institutions, a concise discussion would not adequately address the clinical implications of our findings. The Discussion was intentionally structured to:
-
contextualize the survey responses with current literature,
-
highlight areas of ongoing controversy,
-
emphasize where evidence is lacking and why heterogeneity persists.
Reducing the Discussion would risk oversimplifying interpretation of the survey data and could lead to misrepresentation of its significance. For these reasons, and to preserve scientific rigor and clarity for the reader, we prefer to maintain the current level of detail. We appreciate the reviewer’s perspective, but after careful consideration, we believe that the Discussion in its present form best serves the goals of the manuscript.
-
- We have strengthened the concluding sentence in the Limitations section and added a cautionary interpretation.
-
Thank you for the suggestion. We fully agree that visual representation is valuable, especially in comparative studies. However, in this particular dataset, figures such as a forest plot or regional geographic maps would not add additional scientific value.
-
The survey was descriptive and not outcome-weighted; subgroup comparisons lack sufficient statistical power to support forest plot representation.
-
The geographic distribution of centers was intentionally anonymized as part of the survey design to prevent institutional identification. Adding a map could unintentionally compromise anonymity or imply geographic variability that cannot be statistically supported with the available sample.
To avoid misinterpretation, we believe it is more appropriate to present the results numerically in the tables rather than in visual form
-
Reviewer 2 Report
Comments and Suggestions for Authors
This is an interesting analysis of a multicenter, international survey on pancreaticojejunal anastomosis after PD. The authors reached 122 centers, mostly European and mostly low-volume. The study is well-designed and clearly described. The authors included an appropriate discussion and identified the weak points. I have some minor issues which in my opinion require attention:
- The authors cite a few works on pancreaticojejunal anastomoses. It might improve the manuscript to include more recent studies, e.g., Olakowski M, Grudzińska E, Mrowiec S. Pancreaticojejunostomy-a review of modern techniques. Langenbecks Arch Surg. 2020 Feb;405(1):13-22. doi: 10.1007/s00423-020-01855-6. Epub 2020 Jan 23. PMID: 31975148
- The authors state „Minimally invasive methods, laparoscopic and robotic, were increasingly employed, while open surgery remained more frequent in a notable minority of centers—reflecting the transitional status of PD practice worldwide”. This does not seem true, while only 39.3% of the centers use exclusively open surgery, and the minimally invasive techniques are definitely on the rise, still, in the centers applying these techniques only 21% apply robotic surgery in >50% of the PDs, and only 4.9% apply laparoscopy in >50% of the PDs. This means that, still, open PD is more frequent in most (if not all) centers.
- In the Discussion, the authors present short information on abdominal drainage and somatostatin administration, however, these topics were not covered in the study survey, which should be clearly stated in the paragraph. In the lines 152-155 the Authors indicate the POPF risk factors, the addition of postoperative acute pancreatitis would be advisable as it is directly connected to the POPF formation ( as in Kühlbrey CM, Samiei N, Sick O, Makowiec F, Hopt UT, Wittel UA. Pancreatitis After Pancreatoduodenectomy Predicts Clinically Relevant Postoperative Pancreatic Fistula. J Gastrointest Surg. 2017 Feb;21(2):330-338. doi: 10.1007/s11605-016-3305-x. Epub 2016 Nov 28. PMID: 27896656.).
- As the volume of the centres proved to be the most important factor in POPF incidence, it would be interesting to separately show and discuss the data from these centres to inform the reader which methods are mostly chosen in these centres.
- Conclusions: the first paragraph „Postoperative pancreatic fistula remains the most challenging complication following pancreaticoduodenectomy. Despite continued technical refinement, no single anastomotic technique or device has been unequivocally shown to be superior in preventing POPF.” is not a conclusion of the study, which did not aim at finding the most challenging complications or assessing the superiority of any anastomosis.
Author Response
Comments 2:
This is an interesting analysis of a multicenter, international survey on pancreaticojejunal anastomosis after PD. The authors reached 122 centers, mostly European and mostly low-volume. The study is well-designed and clearly described. The authors included an appropriate discussion and identified the weak points. I have some minor issues which in my opinion require attention:
- The authors cite a few works on pancreaticojejunal anastomoses. It might improve the manuscript to include more recent studies, e.g., Olakowski M, Grudzińska E, Mrowiec S. Pancreaticojejunostomy-a review of modern techniques. Langenbecks Arch Surg. 2020 Feb;405(1):13-22. doi: 10.1007/s00423-020-01855-6. Epub 2020 Jan 23. PMID: 31975148
- The authors state „Minimally invasive methods, laparoscopic and robotic, were increasingly employed, while open surgery remained more frequent in a notable minority of centers—reflecting the transitional status of PD practice worldwide”. This does not seem true, while only 39.3% of the centers use exclusively open surgery, and the minimally invasive techniques are definitely on the rise, still, in the centers applying these techniques only 21% apply robotic surgery in >50% of the PDs, and only 4.9% apply laparoscopy in >50% of the PDs. This means that, still, open PD is more frequent in most (if not all) centers.
- In the Discussion, the authors present short information on abdominal drainage and somatostatin administration, however, these topics were not covered in the study survey, which should be clearly stated in the paragraph. In the lines 152-155 the Authors indicate the POPF risk factors, the addition of postoperative acute pancreatitis would be advisable as it is directly connected to the POPF formation ( as in Kühlbrey CM, Samiei N, Sick O, Makowiec F, Hopt UT, Wittel UA. Pancreatitis After Pancreatoduodenectomy Predicts Clinically Relevant Postoperative Pancreatic Fistula. J Gastrointest Surg. 2017 Feb;21(2):330-338. doi: 10.1007/s11605-016-3305-x. Epub 2016 Nov 28. PMID: 27896656.).
- As the volume of the centres proved to be the most important factor in POPF incidence, it would be interesting to separately show and discuss the data from these centres to inform the reader which methods are mostly chosen in these centres.
- Conclusions: the first paragraph „Postoperative pancreatic fistula remains the most challenging complication following pancreaticoduodenectomy. Despite continued technical refinement, no single anastomotic technique or device has been unequivocally shown to be superior in preventing POPF.” is not a conclusion of the study, which did not aim at finding the most challenging complications or assessing the superiority of any anastomosis.
Response 2:
We thank the reviewer for the thoughtful and constructive comments, which improved our manuscript. Below we address each point and describe changes made.
1. We thank the reviewer for the suggestion. We have added the recommended citation and a brief reference to place contemporary PJ techniques in context
2. We agree. We have revised the text to accurately reflect the survey data, clarifying that, although MIS adoption is increasing, open PD remains the most frequent approach in many centers.
3. We have clarified in the Discussion that drainage strategies and routine somatostatin use, such as ERAS protocols, were not included in the questionnaire; the brief discussion remains only for context and is explicitly noted as such
4. Thank you, we have added a sentence referencing Kühlbrey et al. (2017) and noting postoperative pancreatitis as an established predictor of POPF
5. We have added a focused descriptive paragraph in the Results summarizing the subset of centers performing ≥50 PD/year (n = 30). This paragraph details technique preferences, stent usage, and POPF rates in this subgroup, and we discuss these findings in the Discussion
6. We have revised the opening of the Conclusions to align with the scope of the study and its principal findings, emphasizing that institutional volume and organizational factors correlated with lower grade C POPF rates in our survey cohort. We appreciate the reviewer’s positive appraisal of the overall study design and clarity. The manuscript has been updated accordingly; tracked changes are provided in the revised submission. We hope these revisions address the concerns and improve the clarity and utility of our work.
Sincerely,
Dr. S. Caringi, on behalf of all authors
Reviewer 3 Report
Comments and Suggestions for Authors
Dear authors:
Your manuscript is interesting to know the real life of POPF. The main problem is that the accuracy of data (survey data are not audited).
Comments:
- Introduction I think is too short please explain more deeply the POPF related to technical failure and surgical factors associated to POPF
- Methods: explain how you select centers and how do you loook for duplicates. The re is a sentence with letter bigger than others
- Results: Please include median of PD performed and some percentages like less than 20, 20-50 more than 50. Please more information about POPF rate not only >50 vs <50 Please try to imporve statistical analysis
- Discussion: interesting but you have not asked the relevant topics about you talked: drain, somatostatin, ERAS,...
Author Response
Comments 3:
Dear authors:
Your manuscript is interesting to know the real life of POPF. The main problem is that the accuracy of data (survey data are not audited).
Comments:
- Introduction I think is too short please explain more deeply the POPF related to technical failure and surgical factors associated to POPF
- Methods: explain how you select centers and how do you loook for duplicates. The re is a sentence with letter bigger than others
- Results: Please include median of PD performed and some percentages like less than 20, 20-50 more than 50. Please more information about POPF rate not only >50 vs <50 Please try to imporve statistical analysis
- Discussion: interesting but you have not asked the relevant topics about you talked: drain, somatostatin, ERAS,..
Response 3:
- We agree. The Introduction was expanded to better describe how technical aspects of pancreaticojejunal anastomosis and surgeon-related factors contribute to POPF development.
- Regarding invitation is already clarified. We have added how duplicate submissions were prevented.
-
Thank you for noticing. The formatting issue has been corrected.
-
Thank you for this suggestion. We appreciate the interest in additional subgroup analyses and volume stratification. However, we respectfully chose not to modify the current volume categories nor add median values, for the following methodological reasons:
-
The survey was designed with predefined volume categories, based on commonly accepted thresholds in the literature (<50 vs. ≥50 PD/year), to align with previous international studies on PD centralization and outcomes.
-
Further subdivision of centers into <20, 20–50, and >50 PD/year would reduce the number of centers per stratum, resulting in insufficient statistical power and making comparisons unreliable.
-
Given the anonymous nature of the survey and the limited number of high-volume centers, additional stratification could risk indirect identifiability of individual units.
To ensure methodological consistency and preserve data anonymity, we maintained the original categorization (<50 vs. ≥50 PD/year), which reflects the study design and supports the main objective of evaluating practice variation rather than outcome-stratified performance
-
- We agree. Drain management, somatostatin, and ERAS pathways are discussed only to contextualize results. We clarified explicitly that these variables were not collected in the survey
We appreciate the reviewer’s helpful comments, which improved the clarity and completeness of the manuscript. All changes are tracked in the revised version.
Round 2
Reviewer 3 Report
Comments and Suggestions for Authors
The manuscript has improved.
Only one comment this new sentence has been included twice
No pilot testing or formal validation of the questionnaire was performed before dissemination, which may affect the reliability and interpretability of the responses.
Author Response
Comments 1:
The manuscript has improved.
Only one comment this new sentence has been included twice
No pilot testing or formal validation of the questionnaire was performed before dissemination, which may affect the reliability and interpretability of the responses
Response 1:
Dear reviewer, thank you for your observation. I removed one of them. Kind regards.